# Individually Tailored Remote Physiotherapy Program Improves Participation and Autonomy in Activities of Everyday Life along with Exercise Capacity, Self-Efficacy, and Low-Moderate Physical Activity in Patients with Pulmonary Arterial Hypertension: A Randomized Controlled Study

**DOI:** 10.3390/medicina58050662

**Published:** 2022-05-14

**Authors:** Līna Butāne, Liene Spilva-Ekerte, Andris Skride, Daina Šmite

**Affiliations:** 1Faculty of Rehabilitation, Riga Stradins University, LV-1007 Riga, Latvia; liene.spilva@gmail.com (L.S.-E.); daina.smite@rsu.lv (D.Š.); 2Department of Rehabilitation, Pauls Stradins Clinical University Hospital, LV-1012 Riga, Latvia; 3Faculty of Medicine, Department of Internal Diseases, Rigas Stradins University, LV-1007 Riga, Latvia; andris.skride@rsu.lv; 4Department of Cardiology, Pauls Stradins Clinical University Hospital, LV-1012 Riga, Latvia

**Keywords:** chronic disease, pulmonary arterial hypertension, physiotherapy, individualized, interdisciplinary, quality of life, participation, exercise, physical activity, home-based, self-efficacy

## Abstract

*Background and Objectives:* Pulmonary arterial hypertension (PAH) is a rare, chronic, progressive, and life-threatening disease; however, the appropriate target medical treatment today allows patients with PAH not only to survive but also to live a relatively normal life. However, patients face the challenge of adapting and maintaining a good quality of life, thus it is important to consider complex interventions related not only to medical treatment. *Methods:* This was a prospective, randomized, controlled, single-blind study. Twenty-one (21) patients diagnosed with PAH were included and randomly assigned to training or control group. All participants continued target medical therapy. Furthermore, TG underwent the individually tailored 12-week remote physiotherapy program. As a primary outcome measurement, the Impact on Participation and Autonomy Questionnaire (IPA) was used. Secondary outcome included aerobic capacity (6MWT), accelerometery and general self-efficacy (GSE). Data were collected at baseline, after 12 weeks and at follow-up 6 months after the beginning of the intervention. *Results:* A significant difference between the groups was found in the follow-up assessment on three of the four IPA subscales analyzed, AO, RF, and AI. The total IPA score decreased significantly in TG after the program, indicating an improved participation. In addition, in TG a significant increase in 6MWT results, daily time in low- or moderate-intensity physical activities, and GSE was observed. *Conclusions:* In summary, the individually tailored physiotherapy program investigated added to stable target medical therapy in patients with PAH encourages improvement and prevents possible deterioration of perceived participation of patients in activities of their everyday life in the context of one’s health condition in the long term, along with improved exercise capacity and increased time spent in low- or moderate-intensity physical activities. Future studies are needed to develop and evaluate long-term intervention to support patients living with this rare, chronic, and life-threatening disease.

## 1. Introduction

Pulmonary arterial hypertension (PAH) is a serious disease characterized by an increase in pulmonary artery pressure and pulmonary vascular resistance, which ultimately leads to right ventricular overload and failure, clinically manifested as exertional dyspnea, fatigue, dizziness, and episodes of syncope, which substantially limit the physical capacity of patients [1,2]. PAH is a rare disease with a prevalence of 11 to 26 cases per million adults [2]. It is a chronic, progressive and life-threatening disease, nevertheless, the appropriate target medical treatment today allows patients with PAH not only to survive, but also to live a relatively normal life by adjusting their lifestyle and establishing the optimal adaptation. However, adults living with PAH acknowledge many uncertainties, feel unsecure and isolated, and face the challenge of adapting and maintaining a good quality of life (QoL) [3]. To minimize the impact of PAH on a patient’s life, it is important to consider an interdisciplinary treatment approach with complex interventions related not only to the medical treatment of the underlying disease [4].

In the 6th World Symposium on Pulmonary Hypertension (Nice, 2018) first time the section “Patient Perspectives on Pulmonary Hypertension” was held to involve the representatives of patients to discuss the needs and possibilities of PAH treatment from the point of view of patients. In this session, disease-specific health-related quality of life (HRQoL) outcomes were highlighted as relevant and important results in clinical trials, and it was recommended to integrate these measurements into daily clinical practice [5]. The multidimensionality of the QoL concept embraces the subjective perspective of the individual, perception and meaning, about own life that is disclosed only in a particular context—environmental, social, and cultural. Persons living with chronic diseases desire to live a life free from disease, which is possible if one feels the ability to make their own choices and decisions, perceives the ability to self-care based on their own competence, and feels a sense of belonging [6]. At the same time, along with efforts to achieve self-control over one’s conditions and their treatment, patients strive to reclaim their identity and live a meaningful life [6,7].

An individual’s sense of well-being is shaped by the perceived ability to decide how to live his own life, the idea of decisional autonomy [8]. An individual’s perceived decisional autonomy could be considered a fundamental element of participation, understood as the ability to choose the way one participates in everyday life activities and makes possible the realization of his/her own project of life or, in other words, the achievement of self-realization by building his own meaningful life that expresses his unique identity [9].

Incorporating notions of autonomy and participation in a conceptual understanding of quality of life in relation to an individual’s health condition and disability, an individualized approach becomes possible in the evaluation of patients and the formulation of relevant treatment outcomes. Designing our research project on the individually tailored physiotherapy program in patients with PAH in addition to stable target medical therapy, to include the comprehensive possible benefits of the intervention, as outcome measures included objective functional assessment and patient-reported measure on the perceived opportunity to participate in activities of everyday life in one’s desired way. The purpose of this work is to investigate the effects of the individually tailored physiotherapy program added to stable target medical therapy in PAH patients on perceived participation in activities of everyday life in the context of one’s health condition, along with perceived self-efficacy, exercise capacity and the level of daily physical activities.

## 2. Materials and Methods

### 2.1. Study Population

We recruited patients from the Latvian PH registry with PAH diagnosed by right heart catheterization in two steps. After evaluating the registry data, all relevant PAH patients were selected and then the study selection criteria were applied by a cardiologist specialized in PH. The inclusion criteria for this study were: PAH (idiopathic or connective tissue disease), NYHA functional class II-III, age > 18 years, clinically stable and on optimized medical target therapy for at least 3 months prior to entering the study. At the second stage, the initially selected patients were contacted by phone and invited to the University Clinic for on-site screening. We obtained informed consent from all eligible patients prior to their participation. The University Clinical Research Ethics Committee approved the study protocol (No. 3/08.09.2018), which was also in accordance with the Declaration of Helsinki.

### 2.2. Study Protocol

This is a prospective, randomized, controlled, and single-blind study. We randomly assigned the patients using block randomization to either the training group (TG) or the control group (CG). The study was carried out in collaboration with Pauls Stradins Clinical University Hospital, Latvian Association of PH patients, and Riga Stradins University. We evaluated all participants at baseline, after 12 weeks and 24 weeks after the start (follow-up). Blinded assessor conducted assessments in an on-site visit at the University Clinic, additionally accelerometer data with MOX sensor from 7 consecutive days were obtained. All participants continued target medical therapy under the supervision of the particular study cardiologist specialized in PH. Furthermore, TG underwent an individually tailored 12-week remote physiotherapy program supervised by the particular study physiotherapist specialized in cardio-pulmonology. Both groups received recommendations on daily physical activities at the second assessment (after 12 weeks).

### 2.3. Intervention

The individually tailored 12-week remote supervised home-based physiotherapy program was developed specifically for patients with PAH who receive stable target medical therapy. We tested it in our pilot study, affirming feasibility and safety [10], and deriving notions for further adaptation, therefore, coming to this final program. The program consists of physical exercises, relaxation, self-control enhancement, education, and supervision, as we described in our previous publication [11] and illustrated in Figure 1.

The exercises were adapted individually and clearly explained for each patient. The included training modalities had the following characteristics:(I)Aerobic activity: 20 to 40 min of cyclic activity, three times a week. The type of activity was selected individually (e.g., walking, bicycle ergometer). Intensity: perceived exertion as 5–6 on the 10 point Borg scale and sustained SpO_2_ or a decrease of no more than 5% from baseline. Progression was based on individual tolerability and was provided by an increase in training duration and limited by ‘alarm signs’: maximum heart rate of more than 120 bpm, decrease in SpO_2_ to 85%; perceived exertion as very hard (>6 on the Borg scale) and subjective symptoms of exercise intolerance (severe dyspnea or fatigue, dizziness, pain, etc.).(II)Strength training: five to six resistance exercises (involving upper or lower limbs) with 5 to 10 repetitions in each set, twice a week; performed using the person’s own body weight or low weights (dumbbells or water bottles (0.5–1 kg)).(III)Inspiratory muscle training: with PHILIPS Threshold IMT breathing trainer, five times a week. Progression: repetitions from 3 × 3 to 3 × 7 in each set; resistance from 30% to 65–70% from max.(IV)Relaxation: five times a week. Includes breathing techniques (diaphragmatic breathing; slow breathing; pursed lip breathing; breathing pattern perception and awareness) and elements of progressive neuromuscular relaxation and body awareness.

Self-control included monitoring of heart rate, SpO_2_, perceived exertion on the Borg scale, subjective clinical symptoms, recognition of alarm and warning signs with appropriate action that was initially discussed. To improve compliance, patients were required to fill out a daily study diary. Each participant in the training group received a paper or online format diary and pulse oximeter (Beurer PO 60) to ensure self-control measures.

The educational and motivational elements provided by a physiotherapist (both in verbal and visual handout material) were part of the program. Education included information on the benefits of exercises, relaxation, and optimal self-control, possible adverse events of exercise, options for managing activities of daily life, as well as self-management strategies to cope with the exacerbation of disease symptoms or stress situations.

The program included an initial face-to-face outpatient session with the physiotherapist in the University Clinic to individually adjust the program, prepare, and train to use equipment, as well as to try one’s hand at each program element. To promote sufficient adherence and develop sustained behavior changes, an individualized approach was highlighted. Both intensity and mode of exercises were individually adapted for each participant to integrate them into their daily life (for example, a particular day regime or recreational activities), home environment, and individual preferences and possibilities were discussed. Education and support for the improvement of self-control was adapted for particular baseline comprehension and skills. Supervision was carried out with a paper or online diary in Google Docs at the choice of the participant, and telephone conversations at least once a week. In addition, support and encouragement were important aspects of communication during the whole program. To maximize clinical safety, a second visit on site was arranged for reevaluation and program adjustment at the University Clinic for each participant in the TG 4 weeks after starting the program. Furthermore, to recognize adverse events, BNP and CRO levels were evaluated during the intervention period. The program was led by the specialized physiotherapist in close collaboration with the specialized cardiologist.

### 2.4. Outcome Measures

#### 2.4.1. Primary Outcome

In order to assess the perceived participation, we choose Impact on Participation and Autonomy Questionnaire (IPA). IPA is developed in the Netherlands and includes the concept of decisional autonomy as an integral component of participation, describing the perceived opportunity to live life the way one wants in the context of one’s health condition or disability [12]. With permission from the authors of the original Dutch version, the English version of the IPA questionnaire in Latvian was translated following the guidelines for cross-cultural adaptation of self-report measures [13]. In the pilot study in a sample of patients with cardiopulmonary diseases, high internal consistency was approved for all subscales (Cronbach’s alpha ranging from 0.85 to 0.94).

The 32 items assess perceived participation across 5 subscales: Autonomy indoors (AI) includes the chances of taking care of yourself, the way one wants (washing, dressing, eating, etc.), getting around the house when and where one wants; Family role (FR) includes the role, tasks and responsibilities within the family, doing tasks around the house and garden, using one’s money; Autonomy outdoors (AO) includes items about frequency of social contacts, possibilities to spend leisure time and get around outdoors when and where one wants, leading the life one wants; Social life and relationships (SL) includes quality of social life, relationships, communication, respect and intimacy, helping and supporting other people; Work and education (WE) includes items about paid and voluntary work, education and training. All items have five possible answers (very good, good, fair, poor and very poor), indicating chance of participating when and how one wants. Nine additional items assess perceived problems and are intended to facilitate clinical decision-making (not included in analysis). Based on IPA manual to the English version [14], the median score for the subscale was calculated and used to analyze differences between the TG and the CG. Additionally, to detect possible change in each particular item to better describe individual gain, total summary score was calculated and transformed as percent were higher score present less chance to participate in everyday life due to health problems and used to analyze change between assessments within group participants.

#### 2.4.2. Secondary Outcomes

To assess exercise capacity, a six-minute walk test (6MWT) was performed according to guidelines [15], with SpO_2_, heart rate and blood pressure monitoring. The distance (m) covered during the test was recorded and the minimal clinically significant difference (>33 m) was determined based on previous studies [16].

To evaluate daily physical activities, data from the accelerometer were obtained. We established cooperation with Maastricht University and used their developed instrument MOX Physical Activity Monitor [17]. The MOX is a smart monitoring platform. The sensor was worn on the thigh to accurately measure the level of physical activity such as postures and movements during daily living activities [18]. The sensor was applied and worn for seven consecutive days at all times, including during sleep, bathing, or showering. Participants were encouraged to participate in their routine activities while completing the assessment. On completion, the data were downloaded using MOX software to analyze daily time in each of the physical activity levels (sedentary, standing, low intensity, moderate intensity and high intensity activities) as calculated by software taking into account the registered time unit of acceleration measurement, filter (including lowest sensed frequency) and sensor position based on validated algorithms [18]. In the results, the time spent in each physical activity level expressed as percentage of the total awake time is presented.

To assess perceived self-efficacy, the General Self-efficacy Scale (GSE) was used. GSE is a self-reported questionnaire that includes 10 items to assess a general sense of perceived self-efficacy, and the scale was created with the aim of predicting coping with daily problems and adaptation after experiencing all kinds of stressful life events. Perceived self-efficacy is an operational construct related to subsequent behavior and therefore is relevant for clinical practice and behavior change [19]. We used Latvian translation [20], which approved high internal consistency (Cronbach’s alpha 0.93) and interclass correlation (R = 0.94) in a sample of patient with cardiopulmonary disease. Responses in this scale are made on 4-point scale, and a final composite score in a range from 10 to 40 is obtained, and in analysis we used it as transformed percent were higher score represents more perceived self-efficacy.

### 2.5. Data Analysis

Data were analyzed using IBM SPSS Statistics (v. 23.0). We choose mathematical methods of statistics stepwise, based on small group recommendations [21]. Since IPA median scores for subscales were categorical data, a nonparametric analysis was performed by Mann–Whitney U test to examine the difference between groups. The IPA total transformed percent and secondary outcome data were continuous. For these data, at first the normality assumption in all measures was justified by probability plots recommended to use in small sample sizes [21,22]. Thus we proceeded to evaluate the equal variance assumption by examining the ratio of the largest and smallest variances. Based on the rule of thumb, we concluded that the equal variance assumption is reasonable for data of 6MWT results, thus the two-sample *t*-test was used to compare the means in both, i.e., training and control groups at each assessment. In turn, the equal variance assumption was not reasonable for data from accelerometry, IPA total score and GSE results, therefore, Welch’s extensions to t test assuming unequal variances (heteroscedastic) were performed, respectively [21]. To analyze changes in means within each group between baseline, after 12 weeks and follow-up assessments, a paired two-sample t test was performed. α level 0.05 was chosen, therefore, the results as statistically significant were determined if *p* < 0.05. For repeated measures, the significance values were adjusted using the Bonferroni correction and set α level 0.025. To measure the effect size for the results of the t test Cohen’s d (d) was calculated and its thresholds were interpreted as small (0.2), medium (0.5), and large (0.8) effect [23]. Additionally to measure the effect size for the results of the Mann–Whitney U test, the glass rank biserial coefficient (rg) was performed and its thresholds were interpreted as small (<0.3), medium (0.3–0.5), and large (>0.5) effect [24]. For statistically significant results (*p* < 0.05) post hoc statistical power was calculated using the G-Power software according to the values and a power at least 80% (1 − β ≥ 0.8) was assumed as appropriate to control 1-β error [23].

## 3. Results

### 3.1. Characteristics of the Participants at Baseline

Twenty-one (21) patients diagnosed with PAH were included and randomly assigned to TG or CG (see Figure 2). Detailed characteristics of the participants are shown in Table 1. No significant differences were found between the groups in baseline demographic or clinical characteristics.

### 3.2. Adherence and Adverse Effects

All TG patients demonstrated high adherence to the program, reaching satisfactory performance similar to our pilot study [10]. No adverse events defined as clinical worsening or exacerbation, hospitalization, significant increase in the level of BNP or CRO were observed during the course of the training program, confirming the safety of the program.

### 3.3. Primary Outcomes

A total of 12 participants (5 from TG, 7 from CG) did not respond to the statements on the WE subscale; therefore, this subscale was excluded from further analysis.

As shown in Figure 3 a significant difference between the groups was found in the follow-up assessment in three of four IPA subscales analyzed: AO (*p* = 0.01, rg = 0.66, 1 − β = 0.95), FR (*p* = 0.04, rg = 0.55, 1 − β = 0.8), and AI (*p* = 0.04, rg = 0.51, 1 − β = 0.68) based on median scores. In post hoc analysis results of AO and FR approved appropriate statistical power.

As can be seen in Table 2, further analysis showed that the total IPA score significantly decreased in TG after the program: from baseline to 12 weeks assessment (*p* = 0.005, d = 1.1, 1 − β = 0.96), from baseline to follow-up (*p* = 0.004, d = 1.1, 1 − β = 0.97) pointing to improved participation of patients in their everyday life. For the indicated changes, a large effect size was detected, and appropriate statistical power was reached. No significant changes in CG were observed.

### 3.4. Secondary Outcomes

Table 2 provides an overview of the results of secondary outcome measurement tools. An significant increase in 6MWT results in TG after 12 weeks and at follow-up. In contrast, no significant improvement was presented in CG. Significant differences in 6MWT results between the groups at 12 weeks and follow-up were approved. All these results reached appropriate statistical power (1 − β ≥ 0.8). Accelerometry data show a significant reduction in sedentary time from baseline to 12 weeks in both groups, and from baseline to follow-up in TG. No significant differences were observed between the groups in sedentary time. Only in TG a significant increase in low- or moderate-intensity physical activities was observed from baseline to follow-up, and significant difference between groups at follow-up was present in the mentioned activities. Perceived general self-efficacy increased significantly in TG from baseline to 12 weeks. No improvement was present in CG. At follow-up, statistically significant differences between groups were observed, although the results did not reach the appropriate statistical power.

## 4. Discussion

The results reported here confirm that the investigated individually tailored physiotherapy program added to stable medical target therapy in patients with PAH prevents deterioration of participation in everyday life at follow-up on such IPA subscales as AI, FR, and AO, including activities such as walking both home and outdoors when and where one wants, tasks and responsibilities within the family, frequency of social contacts, possibilities to spend leisure time, etc. Additionally, the results of this study indicate a significant improvement in the total IPA score immediately after completion of the program and at follow-up that approves the possibility of the studied treatment approach to encourage the improvement of the perceived opportunity to live life as one wants to in the context of one’s health condition.

This study appears to be the first to find that an interdisciplinary approach to the treatment of patients with PAH could improve and preserve the perceived participation of patients in their everyday life. Furthermore, it was its first study to highlight the analysis of participation by assessing the perceived ability to decide how to live your own life in the context of the health condition of patients with PH, providing a new way of looking at treatment outcomes by incorporating notions of autonomy and participation in a conceptual understanding of HRQoL. Such a way allows more to reflect the experience of patients about living with long-term disease, for example, Hedman et al. (2015) research disclosed the perspective of older people living with chronic diseases who describe abilities or disabilities of less importance than meaningful everyday life [25]. Consequently, promoting more patient-centered approach in the treatment with respect to an individual’s attitudes, social circumstances, personal effectiveness as contributors to participation and well-being in persons living with chronic health condition [26].

Patient-reported HRQoL is recognized as an important and relevant outcome in PAH clinical trials and it is recommended to integrate such tools into daily clinical practice [5]. For example, most studies on physical activity interventions use one of several generic or disease-specific HRQoL measurement tools [27]. In most research, the generic questionnaire SF-36 is used to describe the impact of the health condition on the performance of various activities as perceived limitations of the patient due to the health condition [28]. Zeng et al. (2020) in the systematic review of the effectiveness and safety of exercise training and rehabilitation in PH, included five studies with SF-36 as an outcome measure in the meta-analysis and demonstrated a statistically significant improvement in HRQoL measurements in all domains of the questionnaire, including physical and psycho-emotional aspects [27]. Unfortunately, the commonly applied conceptional sense of HRQoL as a treatment outcome and, accordingly, selected measurement tools substantially reduce the fundamental multidimensional concept of QoL [29], and mainly exclude the reference points of a particular patient in his own everyday life, as well do not disclose either the ‘abilities to act’ or possible opportunities, such as internal and external resources, which becomes particularly relevant in the context of a long-term illness. Furthermore, Krahn et al. (2021) propose a reconsidered definition of health, as “health is the dynamic balance of physical, mental, social, and existential well-being in adapting to the conditions of life and the environment”. Therefore, health is disconnected from limitations caused by chronic diseases or disabilities, and the key issue for an individual to experience good health and purpose in their life is finding ways to adapt to their situations using internal and external resources [30].

One important aspect that emerged from the analysis is that TG participants through comments written on IPA questionnaires described that their improvement in perceived chance to participate in a variety of outdoor activities (the IPA subscale AO) was related to their ability to plan everything, consideration of feeling both physical and emotional as well as actual weather conditions, adherence to medication routine, and other important aspects of optimal adaptation. Conversely, in CG participants significant deterioration was confirmed exactly on the AO subscale. These results mirror data from studies on the perspective of patients with PH that described the need to organize and plan daily life, pay attention to physical and emotional well-being and follow a routine of sleep and diet [4,31]. Besides, Rawling et al. (2020) in the synthesis of qualitative studies on the experience of adult patients living with PH described four main topics, one of which emphasized the transitional nature of PH. Coming to the stage with the aim of stabilizing the disease, maintaining a good quality of life and survival, the person faces the challenge of finding a balance between restrictions, a sense of loss, and an attempt to live a previous life despite the exacerbation of symptoms and possible worsening, while forming a new identity and life [3].

Highlight the approved effectiveness of the studied intervention, the individually tailored physiotherapy program added to stable target medical therapy, in patients with PAH to maintain the opportunity to participate in activities related to frequency of social contacts in follow-up should be done in the face of PAH as a long-term disease. At the same time, in our study, no substantial deterioration or differences between groups were found in the IPA subscale SL that mainly describes the quality of social relationships, which can be explained by the relatively older age of the participants, and it is highly possible that other issues arise in younger patients, as current evidence from qualitative research suggests that quality concerns of social life would be more relevant in younger patients with PAH as well as in patients with less time after confirmation of diagnosis [3]. Recognizing that social participation, for example, meeting friends, volunteer work, club activities, and quality of social relationships, described as sense of belonging, respect, inclusion, are important predictors of well-being in persons living with long-term health problems [32], interventions to support participation are highlighted as relevant part of the interdisciplinary treatment approach. Furthermore, note that studies of the patient’s perspective [3,4] agree that patients with PAH often describe feeling insecure, isolated, and as living with an invisible disease, therefore, to fully eliminate the impact of PAH on a patient’s life, complex interventions in addition to medical treatment of the underlying disease are important, and therefore our treatment approach has the potential to support an individual living with PAH to cope with illness. The observed benefits of the investigated program to improve the perceived chance of participation in activities related to frequency of social contacts in the PAH patients might be due to the change in the perception about their health condition and the improved sense of self-control abilities, as is approved relevant correlations between these factors in other studies [33].

The results of this study indicated a significant improvement in perceived general self-efficacy immediately after the 12-week intervention, indicating better coping with daily problems and better adaptation; however, it should be noted that this improvement was no longer statistically significant in the follow-up assessment, suggesting the need for regular long-term collaboration with healthcare professionals. This is also discussed in the literature data on patients with PH perspective [3,34]. Furthermore, in TG at follow-up a possible clinically important decrease in mean GSE score (Cohen’s d = 0.5) was observed without statistical significance, but indicating a trend of possible worsening of perceived self-efficacy. The results are in agreement with the study by Fors et al. (2018) that approved the effectiveness of a person-centered telephone support intervention to mitigate worsening self-efficacy in patients with chronic obstructive pulmonary disease (COPD) and/or chronic heart failure after 6 months [35], besides the subsequent study by Ali et al. (2021) add that the same intervention was effective to improve task-specific self-efficacy in patients with COPD [36]. We used a similar approach in our program in weekly telephone conversation with each participant to provide support and encouragement; therefore, such or similar intervention could be applied as an optimal solution to provide long-term healthcare service to ensure the sustainability of the results and prevent deterioration of perceived self-efficacy. Ali and colleagues (2021) similarly conclude from their data that person-centered care by telephone could be a successful way to help patients handle their illness [36].

This study provides further evidence on complex exercise programs to improve exercise capacity in patients with PAH who receive stable target medical therapy, confirming a significant improvement in the 6MWT results in TG participants. Waller et al. (2020) review on the effectiveness of various exercise programs in patients with PAH revealed an increase in the mean distance of 6MWT of 40–69 m in 12-week programs (*n* = 5), of which only one was home-based and revealed a 40-m improvement in the distance of 6MWT, thus our results after 12 weeks demonstrate a quite similar improvement (51.8 m), but in follow-up (75.5 m) confirm superiority [37]. Similar results were obtained in the recent pilot study by Wojciuk et al. (2021) that confirmed the long-term efficacy of a 6-month program in improving aerobic capacity (mean 71.38 ± 83.4 m) one year after the initial evaluation [38]. The present findings indicate that the investigated fully home-based program added to target medical therapy is not only a safe intervention in patients with PAH, but also shows an advantage due to its individualized approach, complex nature and integration in daily life, thus leading to long-term improvement in exercise capacity.

In addition, the results of this study reveal an increase in the level of low and moderate intensity daily physical activity in the follow-up assessment in TG participants, pointing to the expected benefit of the program in changing daily habits and health behaviors, in addition to highlighting the need for time to change behaviors. This again encourages thinking about long-term collaboration with health professionals to support and motivate and resonates with the perspective of the patient in meeting the challenges of the adaptation process during PAH treatment [3]. Instead, in CG no increase in dynamic daily physical activity was detected. These results are in line with findings from the Chi et al. (2020) international survey of exercise experience for people living with PH indicating that supervision of exercise programs, psychological support, and specific education are factors that promote activity in the target group, along with uncertainty about the safety and benefits of physical activity, the perceived limitations of symptoms, fear, and anxiety as potential barriers [39].

### Strenghts and Limitations

Some constraints should be considered. First, the small sample size. However, it should be taken into account that PH is a rare disease, in the study only an isolated group were included, and this was a study based on the national PH register. Furthermore, two participants in CG had to be excluded due to exacerbation of the disease or acute COVID-19 infection. Through the careful selection of appropriate statistical tests through the stepwise checking of statistical assumptions along with performed post hoc analysis to detect achieved statistical power, thus control beta error, meaningful conclusions may be drawn. Furthermore, the main results in the post hoc analysis confirmed sufficient statistical power. Second, the current study was unable to describe and analyze perceived participation in work and education activities (IPA subscale WE), due to the small number of answers provided to the relevant statements and that could be explained by the retirement or preretirement age in large part of the participants along with general labor market trends in the country and with weak traditions of volunteering. Furthermore, the restrictions caused by the pandemic, namely forced unemployment, affected some of the participants. In addition, the current study was limited to ensure blinding of participants that could bring bias through patients self-reported reported results.

The beginning of the COVID-19 pandemic during the research process was a challenge for all involved, but since the nature of this program involved remote monitoring and supervision, its implementation was not interrupted, thus indirectly confirming its availability in the case of limited on-site services.

The novelty of our work lies both in the methodology of the intervention with the patient-centred approach emphasizing individual needs, possibilities, and preferences, and in the notion of the perceived opportunity of the patient to participate in his/her daily life activities as he/she wants. As the primary outcome measure in this study, the Latvian version of the IPA survey was used, which has not been used in this target group so far. In our pilot study to test the feasibility and safety of the program [10], we used the SF-36 questionnaire as an HRQoL measurement; hence, we explore an advantage of including participation and its assessment tool, both revealing the process of adaptation and changing focus from limitation to ability and opportunity. We suggest that more studies are needed to explore an interdisciplinary treatment approach since PAH as a chronic disease has the diverse effects on each patient as an individual with respect to their life context.

## 5. Conclusions

In summary, the individually tailored physiotherapy program investigated added to stable target medical therapy in PAH patients encourages improvement and prevents the possible deterioration of the perceived participation of patients in activities of their everyday life in the context of one’s health condition in the long term, along with improved exercise capacity and increased time spent in low or moderate intensity physical activities. The present findings might help promote the use of an interdisciplinary treatment approach in a clinical setting to minimize the impact of PAH on a patient’s life, improving the ability to choose the way one participates in his own everyday activities, thus building his own meaningful life. Future studies are needed to develop and evaluate long-term intervention to support patients living with this rare, chronic, and life-threatening disease.

## Figures and Tables

**Figure 1 medicina-58-00662-f001:**
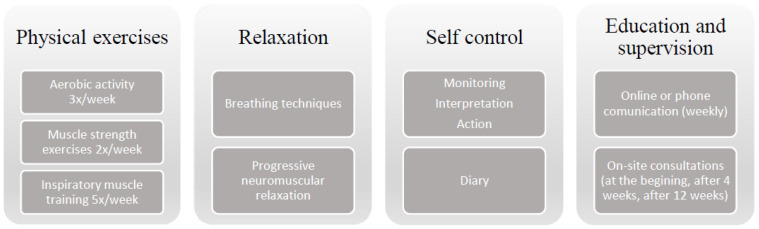
The physiotherapy program content.

**Figure 2 medicina-58-00662-f002:**
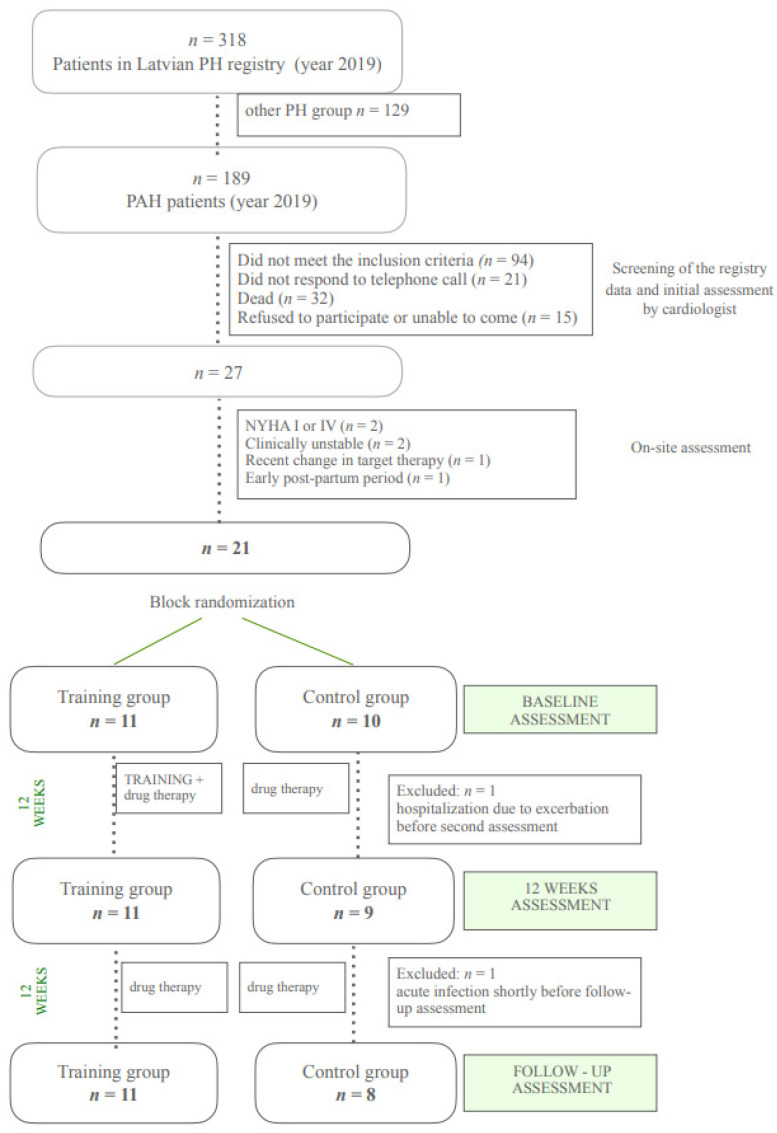
The study flow chart.

**Figure 3 medicina-58-00662-f003:**
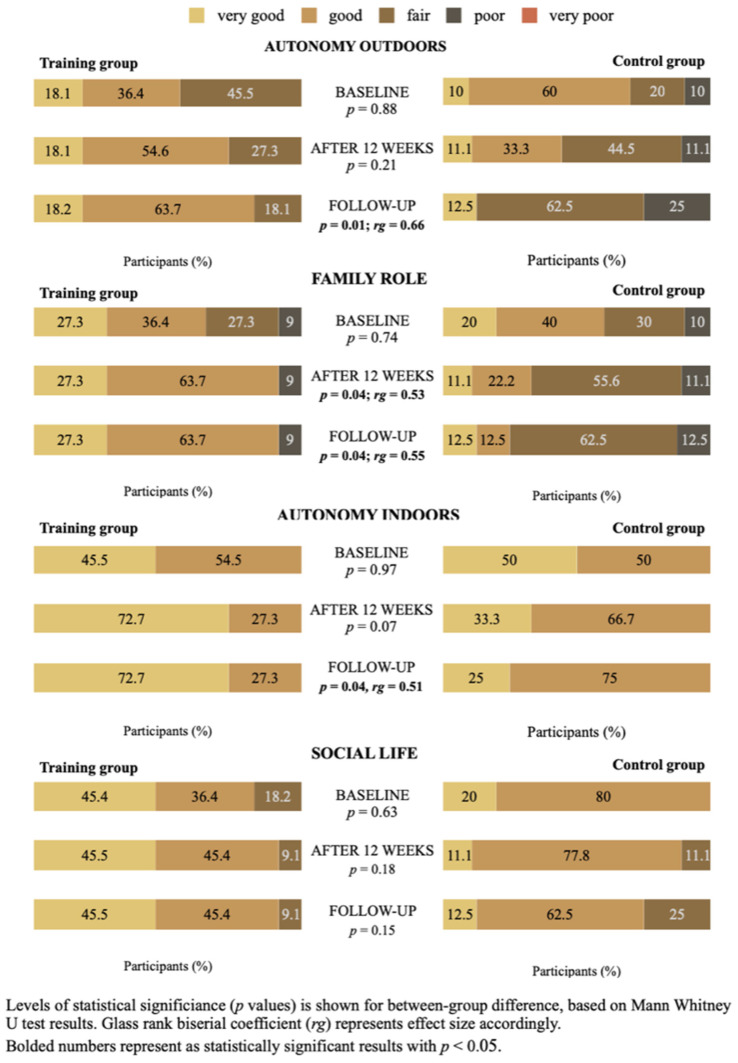
Comparison of IPA sub-scales median score at baseline, after 12 weeks and follow-up assessments in training and control groups.

**Table 1 medicina-58-00662-t001:** Characteristics of participants.

Variables	TG (*n* = 11)	CG (*n* = 10)	Difference between Groups (*p*)
Age (years)	68 (16)	66 (11.5)	0.78
(*n*) < 45	2	1	
45–55	-	-	
55–65	2	2	
65–75	5	5	
>75	2	2	
Gender (*n*) Women/Men	10/1	9/1	
BMI (kg/m^2^)	25.7 (6.7)	26.7 (10.4)	0.41
(*n*) 18.5–24.99	6	5	
25–29.99	2	1	
>30	3	4	
PAH etiology (*n*)			
Idiopathic	6	4	
Connective tissue disease	2	3	
Congenital heart disease	2	2	
PAH target therapy (*n*)			
PDE5 inhibitor	11	10	
ERA	4	5	
Ventavis	1	-	
Spironolactonum	11	9	
Oxygen therapy	-	2	
Co-morbidities (*n*)			
Hypertension	6	5	
Dislipidemia	5	7	
CHF	9	7	
AF	7	5	
			Time since diagnosis (years)
Cardiac catheterization			
mPAP (mmHg)	46 (15.2)	54.5 (20.5)	0.31
PVR (WU)	6.8 (2.1)	7.8 (3.5)	0.57
PAWP (mmHg)	11 (4.5)	11.5 (4.3)	0.72
(*n*) ≤ 12	7	6	
15–13	2	2	
>15	2	2	
Cpc-PH *	2	2	
Echocardiographic data			
TAPSE (mm)	18 (4)	18 (5)	0.93
LVEF (%)	50 (11.5)	50 (2)	0.97
Spirometry			
FVC (% pred)	72 (21.5)	71 (26)	0.54
FEV1 (% pred)	74 (7.5)	76 (26)	0.88
FEV1/FVC (% pred)	90.6 (11.1)	87 (24.1)	0.24

Data are presented as *n* (number of participants) or median (interquartile range, IQR). Difference between the groups tested by Mann–Whitney U test. * Cpc-PH: PAWP > 15 mmHg and DPG ≥ 7 mmHg and/or PVR > 3 WU; three participants with Cpc-PH had congenital heart disease (two from TG, one from CG), but one a connective tissue disease (systemic sclerosis) (from CG). Ratio FEV_1_/FVC less than 65% were observed in two CG participants along with FEV1 <80%. Abbreviations: BMI, body mass index; PAH, pulmonary arterial hypertension; PH, pulmonary hypertension; PDE, phosphodiesterase; ERA, endothelin receptor antagonist; CHF, chronic heart failure; AF, atrial fibrillation; mPAP, mean pulmonary arterial pressure; PAWP, pulmonary arterial wedge pressure; PVR, pulmonary vascular resistance; Cpc-PH, combined postcapillary and precapillary pulmonary hypertension; TAPSE, tricuspid annular plane systolic excursion; LVEF, left ventricular ejection fraction; FVC, forced vital capacity; FEV_1_, forced expiratory volume in one second.

**Table 2 medicina-58-00662-t002:** Effects of physiotherapy program on exercise capacity, daily physical activity level, perceived self-efficacy, participation and autonomy.

Characteristics		Baseline Mean ± SD	After 12 Weeks Mean ± SD	Change within Group *p* Value (Cohen’s d Value)	Follow-Up Mean ± SD	Change within Group *p* Value (Cohen’s d Value)
6MWT results (distance, m)	TG	378.3 ± 124.3	450 ± 114	**0.001** (**1.7**) ******	473.6 ± 118.8	**<0.001 **(**2.1**)** ****
	CG	296.1 ±110.1	290.6 ± 112.2	0.84 (0.2)	302.5 ± 139.7	0.13 (0.2)
Difference between the groups *p* value (Cohen’s d value)		0.12 (0.7)	**0.01** (**1.4**) ******		**0.01** (**1.3**) ******	
Accelerometry results (% from total awake time) SEDENTARY	TG	67.2 ± 8.8	60.7 ± 10.1	**0.01** (**0.9**) ******	58.1 ± 10.1	**0.005** (**1.0**) ******
	CG	75.6 ± 3.6	63.4 ± 9.1	**0.003** (**1.4**) ******	65.2 ± 11.2	**0.04** (**0.9**)
Difference between the groups *p* value (Cohen’s d value)		0.20 (0.9)	0.55 (0.2)		0.17 (0.6)	
STANDING	TG	25.4 ± 6.4	28.7 ± 8.8	**0.03** (**0.8**)	28.8 ± 8.9	0.06 (0.7)
	CG	21.6 ± 5.7	29.8 ± 7.7	**0.03** (**0.9**)	28.8 ± 9.7	**0.03** (**0.7**)
Difference between the groups *p* value (Cohen’s d value)		0.18 (0.6)	0.72 (0.1)		0.97 (0)	
LOW INTENSITY	TG	1.3 ± 0.4	1.6 ± 0.5	**<0.001** (**1.6**) ******	1.8 ± 0.7	**0.002** (**1.2**) ******
	CG	1.0 ± 0.6	1.1 ± 0.4	0.77 (0.3)	0.9 ± 0.4	0.60 (0.2)
Difference between the groups *p* value (Cohen’s d value)		0.23 (0.6)	**0.04** (**1.1**)		**0.005** (**1.6**) ******	
MODERATE INTENSITY	TG	7.1 ± 3.4	8.0 ± 2.4	0.21 (0.4)	9.5 ± 3.5	**0.002** (**1.3**) ******
	CG	4.9 ± 2.8	5.4 ± 2.0	0.67 (0.2)	4.8 ± 1.8	0.19 (0.0)
Difference between the groups *p* value (Cohen’s d value)		0.11 (0.7)	**0.02** (**1.2**)		**0.002** (**1.7**) ******	
HIGH INTENSITY	TG	1.4 ± 1.4	1.3 ± 0.8	0.78 (0.1)	1.7 ± 1.5	0.50 (0.2)
	CG	0.3 ± 0.3	0.4 ± 0.6	0.28 (0.2)	0.3 ± 0.3	0.14 (0)
Difference between the groups *p* value (Cohen’s d value)		**0.03** (**0.8**)	**0.01** (**1.1**)		**0.01** (**0.9**)	
GSE results (transformed %)	TG	64.9 ± 22.7	74.2 ± 17.5	**0.004** (**1.04**) ******	69.4 ± 16.5	0.22 (0.34)
	CG	55.2 ± 25.0	50.7 ± 20.1	0.37 (0.28)	46.3 ± 25.0	0.25 (0.51)
Difference between the groups *p* value (Cohen’s d value)		0.62 (0.24)	**0.014** (**1.21**)		**0.043** (**1.08**)	
IPA total score (transformed %)	TG	24.4 ± 13.0	20.0 ± 10.7	**0.005** (**1.1**) ******	18.6 ± 10.2	**0.004** (**1.1**) ******
	CG	25.6 ± 16.1	30.6 ± 18.0	0.092 (0.6)	31.3 ± 20.2	0.068 (0.8)
Difference between the groups *p* value (Cohen’s d value)		0.98 (0)	0.15 (0.7)		0.14 (0.8)	

To explore the difference between the groups, we used *t*-test for independent samples (with Welch’s extension), but for detecting the change in mean values within each group between baseline and after 12 weeks or baseline and follow-up assessments we performed paired two-sample *t*-test. Bolded numbers represent as statistically significant results with *p* < 0.05 or *p* < 0.025 (with Bonferroni adjustment). ** Results are statistically significant and achieved appropriate power as both *p* < 0.05 and 1 − β ≥ 0.8 was observed. Abbreviations: 6MWT, six-minute walk test; GSE—General Self-Efficacy Scale; IPA—Impact on Participation and Autonomy questionnaire; SD, standard deviation.

## Data Availability

Data are available upon request from the corresponding author.

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
