# Peer review of "Individually Tailored Remote Physiotherapy Program Improves Participation and Autonomy in Activities of Everyday Life along with Exercise Capacity, Self-Efficacy, and Low-Moderate Physical Activity in Patients with Pulmonary Arterial Hypertension: A Randomized Controlled Study"

_medicina, 2022, doi:10.3390/medicina58050662_

Round 1
Reviewer 1 Report
Authors demonstrated the effectiveness of remote physiotherapy program for improved daily-life activity and exercise tolerance in patients with PAH. I thought this remote exercise program was a great challenge to improve the QOL in patients with PAH.
Major comments:
- How did the authors determine the intensity of the home-based exercise in each patient? Did participants undergo cardiopulmonary exercise tests to evaluate the exercise capacity?
- In Table 1, I thought the mean age seemed to be high for PAH patient cohort. PAWP was over 12 mmHg in both groups, indicating the presence of elevated left ventricle end-diastolic pressure (like group 2 PH). BMI in both group was also high, which is atypical for pure IPAH patients. Since the number of patients is limited, authors should demonstrate the demographics and hemodynamics of each patient.
- Parameters of right ventricle and left ventricle functions (measured by echocardiogram or cardiac MRI) should be added in Table 1.
- Since all participants were encouraged to undergo this exercise-associated study, I thought this study had a selection bias.
- This study was a single-blind RCT (patients were not blind), and the outcomes of this study included patient reported outcomes. This was a limitation of this study.
Minor comments:
- In abstract, add the full spelling of TG and CG in line 7.
- Which patients do authors think eligible for this training program? All PAH patients? Patients who had high BMI and/or low respiratory capacity?
Reviewer 2 Report
This interesting and meaningful clinical study provides the effects of the individually tailored physiotherapy program added to stable target medical therapy in PAH patients.
Major comments:
- There are apparent baseline differences between the training group (TG) and the control group (CG), such as age and hemodynamic index. These differences may lead to result bias.
- The sample size is small.
Round 2
Reviewer 1 Report
Authors clearly answered the requests and improved their manuscript.
Author Response
Thank you for response to our answers and improvements on the manuscript.